

# Three-dimensional simulation of stratospheric gravitational separation using the NIES global atmospheric tracer transport model

Dmitry Belikov[1], Satoshi Sugawara[2], Shigeyuki Ishidoya[3], Fumio Hasebe[1], Shamil Maksyutov[4], Shuji Aoki[5], Shinji Morimoto[5], and Takakiyo Nakazawa[5]

[1]Faculty of Environmental Earth Science, Hokkaido University, Sapporo 060-0810, Japan
[2]Miyagi University of Education, Sendai 980-0845, Japan
[3]National Institute of Advanced Industrial Science and Technology (AIST), Tsukuba 305-8569, Japan
[4]National Institute for Environmental Studies, Tsukuba 305-8506, Japan
[5]Center for Atmospheric and Oceanic Studies, Tohoku University, Sendai 980-8578, Japan

**Correspondence:** Dmitry Belikov (dmitry.belikov@ees.hokudai.ac.jp)





**Abstract.** A three-dimensional simulation of gravitational separation, defined as the process of atmospheric molecule separation under gravity according to their molar masses, is performed for the first time in the upper troposphere and lower stratosphere. We analyze distributions of two isotopes with a small difference in molecular mass ($^{13}C^{16}O_2$ ($M_i$ = 45) and $^{12}C^{16}O_2$ ($M_i$ = 44)) simulated by the National Institute for Environmental Studies (NIES) chemical transport model with a parameterization of molecular diffusion. The NIES model employs global reanalysis and an isentropic vertical coordinate and uses optimized $CO_2$ fluxes. This study includes a comparison with measurements recorded by cryogenic balloon-borne samplers in the lower stratosphere and two-dimensional model simulations. The benefits of the NIES TM simulations are discussed. We investigate the processes affecting gravitational separation by a detailed estimation of terms in the molecular diffusion equation. At the same time, we study the age of air derived from the tracer distributions. We find a strong relationship between age of air and gravitational separation for the main climatic zones. The advantages and limitations of using age of air and gravitational separation as indicators of the variability in the stratosphere circulation are discussed.

## 1 Introduction

Chapman and Milne (1920) proposed two different dynamical regimes: the homosphere (perfect mixing) in the lower part of the atmosphere and the heterosphere (diffusive equilibrium), where molecular diffusion leads to a separation of the atmospheric constituents according to their molecular mass. The process of atmospheric molecule separation according to their molar mass under gravity is termed "gravitational separation" (GS). The GS of atmospheric components is recognized as dominant in the atmosphere above a level of about 100 km called the turbopause (Warneck and Williams, 2012). Recently, the existence of GS of the major atmospheric components in the stratosphere was confirmed both experimentally using a precise cryogenic sampler and theoretically by two-dimensional numerical model simulations (Ishidoya et al., 2008, 2013).

The Brewer–Dobson circulation (BDC) is the global circulation in the stratosphere, consisting of air masses that rise across the tropical tropopause, then move poleward and descend into the extratropical troposphere (Butchart, 2014). Currently, the "mean age of air" (mean AoA), defined as the average time that an air parcel has spent in the stratosphere, is perhaps best known for being a proxy for the rate of the stratospheric mean meridional circulation and the BDC. The mean AoA provides an integrated measure of the net effect of all transport mechanisms on the tracers and air mass fluxes between the troposphere and stratosphere (Hall and Plumb, 1994; Waugh and Hall, 2002).

The increase in greenhouse gas abundance is leading to increased radiative forcing and therefore to warming of the troposphere and cooling of the stratosphere (Field et al., 2014). A strengthened BDC under climate change in the middle and lower stratosphere is robustly predicted by various chemistry–climate models (Butchart et al., 2006; Butchart, 2014; Garcia and Randel, 2008; Li et al., 2008; Stiller et al., 2012; Garfinkel et al., 2017). However, this is not in agreement with over 30 years of observations of the age of stratospheric air (Engel et al., 2009; Hegglin et al., 2014; Ray et al., 2014).

The sparse limited observations in the upper troposphere and lower stratosphere (UTLS) mean that chemical transport models (CTMs) are complementary and useful tools for widely diagnosing the BDC and representing the global transport and distribution of long-lived species. CTMs perform relatively well in the UTLS despite resolution issues. Confidence is



high in the ability of models to reproduce many of the features, including the basic dynamics of the stratospheric BDC and the tropospheric baroclinic general circulation in the extratropics, the tropopause inversion layer, the large-scale zonal mean, tropical and extratropical tropopause (Hegglin et al., 2010; Gettelman et al., 2011).

As future changes to the BDC are likely to be complex, a suite of methods, parameters, and tools is necessary to detect these changes. Leedham Elvidge et al. (2018) evaluated the capability of using seven trace gases to estimate stratospheric mean ages. Ishidoya et al. (2013) proposed using GS as an indicator of changes in the atmospheric circulation in the stratosphere. Analyses of GS, in addition to the $CO_2$ and $SF_6$ ages, may be useful for providing information on stratospheric circulation.

Ishidoya et al. (2013) also performed the first simulation of GS using the NCAR two-dimensional (2-D) SOCRATES model (Simulation Of Chemistry, RAdiation, and Transport of Environmentally important Species, (Huang et al., 1998)), which is an interactive chemical, dynamical, and radiative model. The spatial domain of the model extends from the surface to 120 km in altitude. The vertical and horizontal resolutions are 1 km and 5°, respectively. To extend the earlier work and overcome the inherent limitations of the 2-D model, we here present a more quantitative analysis using the National Institute for Environmental Studies (NIES) Eulerian three-dimensional (3-D) transport model (TM). The remainder of this paper is organized as follows. Overviews of the NIES TM, a method for modeling GS, and the simulation setup are provided in Sect. 2. In Sect. 3 we study modeled GS and compare vertical profiles with those observed. Finally, a summary and conclusions are provided in Sect. 4.

## 2 Model and method

For further investigation of the GS process we redesigned and modified the NIES TM, which has previously been used to study the seasonal and inter-annual variability of greenhouse gases (i.e., $CO_2$, $CH_4$ by Belikov et al. (2013b)).

### 2.1 Model

The NIES model is an off-line transport model driven by the Japanese Meteorological Agency Climate Data Assimilation System (JCDAS) datasets (Onogi et al., 2007). It employs a reduced horizontal latitude–longitude grid with a spatial resolution of 2.5°× 2.5°near the equator (Maksyutov and Inoue, 2000) and a flexible hybrid sigma–isentropic ($\sigma$–$\theta$) vertical coordinate, which includes 32 levels from the surface up to 5 hPa (Belikov et al., 2013b).

Over the isentropic part of the grid, which starts from the potential temperature level of 295K and includes the tropopause, the vertical transport follows the seasonally varying climatological diabatic heating rate derived from reanalysis. Following the approach by Hack et al. (1993), transport processes in the planetary boundary layer (height provided by the ECMWF ERA-Interim reanalysis (Dee et al., 2011)) and in the free troposphere are separated with a turbulent diffusivity parametrization. The modified cumulus convection parameterization scheme computes the vertical mass fluxes in a cumulus cell using conserva-tion of moisture derived from a distribution of convective precipitation in the reanalysis dataset (Austin and Houze Jr, 1973; Belikov et al., 2013a). To set cloud top and cloud bottom height a modified Kuo-type parameterization scheme (Grell et al.,




1995) is used. Computation of entrainment and detrainment processes accompanying the transport by convective updrafts and downdrafts is as employed by Tiedtke (1989).

## 2.2 Molecular diffusion

According to Banks and Kockarts (1973), assuming a neutral gas, the equation for the vertical component of the diffusion velocity of gas 1 relative to gas 2 ($w = w_1 - w_2$) in a binary mixture of gas 1 and gas 2 in a gravitational field can be written as

$$w_1 - w_2 = -D_{12}\left[\frac{n^2}{n_1 n_2}\frac{\partial(n_1/n)}{\partial z} + \frac{M_2 - M_1}{M}\frac{1}{p}\frac{\partial p}{\partial z} + \frac{\alpha_T}{T}\frac{\partial T}{\partial z}\right], \tag{1}$$

where $n_1$, $n_2$ are the concentrations of particles 1 and 2, respectively, $n = n_1 + n_2$ is the total concentration of the binary mixture, $T$ is the absolute neutral gas temperature, $p$ is the total pressure, $M_1$, $M_2$ are the masses of particles 1 and 2, $M = (n_1 M_1 + n_2 M_2)/(n_1 + n_2)$ is the mean molecular mass, $D_{12}$ is the molecular diffusion coefficient of gas 1 in gas 2, and $\alpha_T$ the thermal diffusion factor. The three terms in the brackets represent, from left to right, the effects of concentration gradient, pressure gradient, and temperature gradient, respectively, on GS.

If component $i$ is a multicomponent mix of minor constituents, then using the hydrostatic equation, the perfect gas law, and Equation 1, the diffusion velocity $w_i$ for the $i$th minor constituent can be written in time-independent form (for more details see pp. 33–34 in Banks and Kockarts (1973)):

$$w_i = -D_i\left[\frac{1}{n_i}\frac{\partial n_i}{\partial z} + \frac{1}{H_i} + (1 + \alpha_{Ti})\frac{1}{T}\frac{\partial T}{\partial z}\right], \tag{2}$$

where $H_i = kT/M_i g$ is the scale height, $k = 1.38 \times 10^{-23}$ J K$^{-1}$ is the Boltzmann constant, $g = 9.81$ m s$^{-2}$ is the standard acceleration due to gravity for Earth, and $D_i$ is the molecular diffusion coefficient. Here also $n_i$, and $\alpha_{Ti}$ are the number density and the thermal diffusion factor for species $i$, respectively.

Similar to the SOCRATES model the vertical component of velocity is converted to the vertical component of molecular diffusion flux of a minor constituent $i$ relative to air (Huang et al., 1998):

$$f_i = -D_i\left[\frac{\partial n_i}{\partial z} + \frac{n_i}{H_i} + (1 + \alpha_{Ti})\frac{n_i}{T}\frac{\partial T}{\partial z}\right]. \tag{3}$$

The molecular diffusion coefficient is estimated from kinetic gas theory by Banks and Kockarts (1973) to be:

$$D_i[\text{cm}^2/\text{s}] = 1.52 \times 10^{18}\sqrt{\frac{1}{M_i} + \frac{1}{M}} \times \frac{\sqrt{T}}{N}, \tag{4}$$

where $M_i$ and $M$ are the mass of the minor constituent $i$ and the mean molecular mass in atomic mass units, respectively, and $N$ is the number density of air.





To derive a diffusive flux formulation consistent with the NIES model transport equation, the number density $n_i$ is substituted by the mixing ratio $C_i = n_i/n$:

$$f_i = -D_i \times N\left[\frac{\partial C_i}{\partial z} + \left(\frac{1}{H_i} - \frac{1}{H}\right)C_i + \alpha_{T_i}\frac{C_i}{T}\frac{\partial T}{\partial z}\right]. \qquad (5)$$

Here $H$ is the atmospheric scale height, and $\alpha_{T_i}$ is assumed to be zero since the thermal diffusion effect would be of no importance in the stratosphere (Ishidoya et al., 2013). The derived flux was added to the standard transport formulation for each species.

## 2.3 Simulation setup

The study of GS requires considering two isotopes of atmospheric tracers with a small difference in molecular mass. Following the setup for the SOCRATES baseline-atmosphere run (Ishidoya et al., 2013), we calculated the distributions of $^{13}\text{C}^{16}\text{O}_2$ ($M_i$ = 45) and $^{12}\text{C}^{16}\text{O}_2$ ($M_i$ = 44). Firstly, a 20-year spin-up calculation with $CO_2$ using biospheric and oceanic fluxes only, and then a 29-year (1988–2016) simulation with total $CO_2$ fluxes (biospheric, oceanic and fossil fuel) were performed. The fluxes used were obtained with the GELCA-EOF (Global Eulerian–Lagrangian Coupled Atmospheric model with Empirical Orthogonal Function) inverse modelling scheme (Zhuravlev et al., 2013). This set of fluxes reproduces realistic time and spatial distributions of $CO_2$ mixing ratio with strong seasonal variations in the Northern Hemisphere (NH) and weak variations in the Southern Hemisphere (SH). The period 1988–2014 is covered by the original JCDAS, which is extended by JRA-55 (an updated version of the Japanese reanalysis) remapped to the same horizontal and vertical grid, as JCDAS production discontinued after 2014. Use of JRA-55 for the whole simulated period is preferable; however, model redevelopment is required to take full advantage of the improved vertical and horizontal resolutions.

The $<\delta>$ value, a measure of the GS, is defined as the isotopic ratio of the $CO_2$:

$$<\delta> = \delta(^{13}\text{C}^{16}\text{O}_2) = \frac{\left[n(^{13}\text{C}^{16}\text{O}_2)/n(^{12}\text{C}^{16}\text{O}_2)\right]_{strat}}{\left[n(^{13}\text{C}^{16}\text{O}_2)/n(^{12}\text{C}^{16}\text{O}_2)\right]_{\text{trop}}} - 1, \qquad (6)$$

where subscripts *strat* and *trop* denote the tropospheric and stratospheric values, respectively. As the tropospheric value *trop* we selected the model tracer concentration from the third level, which corresponds to the lower boundary of the free troposphere. The tropospheric $<\delta>$ variations are very small and negligible compared with those in the stratosphere.

$CO_2$ is a useful tracer of atmospheric dynamics and transport due to its long lifetime. It is chemically inert in the free troposphere and has only a small stratospheric source (up to 1 ppm) from methane oxidation (Boucher et al., 2009). Sufficiently accurate estimation of emissions and sinks, together with knowledge of their trend in combination with the good performance of the NIES model in simulating greenhouse gases make the selection of $CO_2$ appropriate for this study.



## 3 Results and discussion

### 3.1 Zonal mean distribution

The zonal mean distribution of $CO_2$ in the upper part of the atmosphere is driven by the large-scale transport processes: fast quasi-isentropic mixing is combined with upwelling in the tropics and downwelling in the extratropical lowermost stratosphere.

In the troposphere, vertical mixing is well developed. With height, the dynamic characteristics weaken, and the mass flux due to molecular diffusion increases (Eq. 3). At a certain level near the tropopause, vertical mixing is no longer able to suppress diffusion and the $<\delta>$ value becomes nonzero.

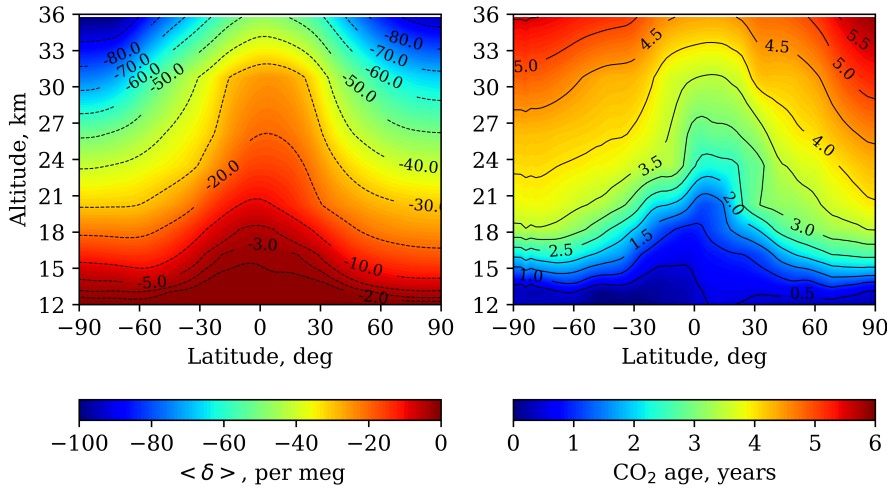

**Figure 1.** Annual mean altitude–latitude distributions of $<\delta>$ value [per meg] (left) and $CO_2$ age [years] (right) calculated using NIES TM. The unit [per meg] is typically used in isotope analysis and is the same as $[10^{-3}$ permil].

Together with the $<\delta>$ value we analyze the AoA (Fig. 1), which is calculated using the lag method (Waugh et al., 2013). The age $a(x)$ at a particular location $x$, is defined as the time since the mixing ratio in the "source region", $C_0$, equaled the mixing ratio at that location $C(x)$,

$$a(x) = \frac{C(x) - C_0}{\frac{dC_0}{dt}}. \tag{7}$$

To calculate the growth rate $dC_0/dt$ we use a linear fit of mixing ratio $C_0$, which is an aggregated combination from two remote sites: Samoa (14.24°S, 170.56°W) and Mauna Loa (19.55°N, 155.58°W) (GLOBALVIEW-CO2, 2013). A comparison with the results from the boundary impulse response method (Li et al., 2012) will be discussed in a separate paper under preparation.

The processes creating the deformation of the zonal mean cross-sections of the GS and the AoA are similar here: the tropical upwelling pumps in tropospheric air and stretches the parameter profile upward (Fig. 1). The NIES TM results shows strong subtropical mixing barriers in both hemispheres compared with the SOCRATES model.



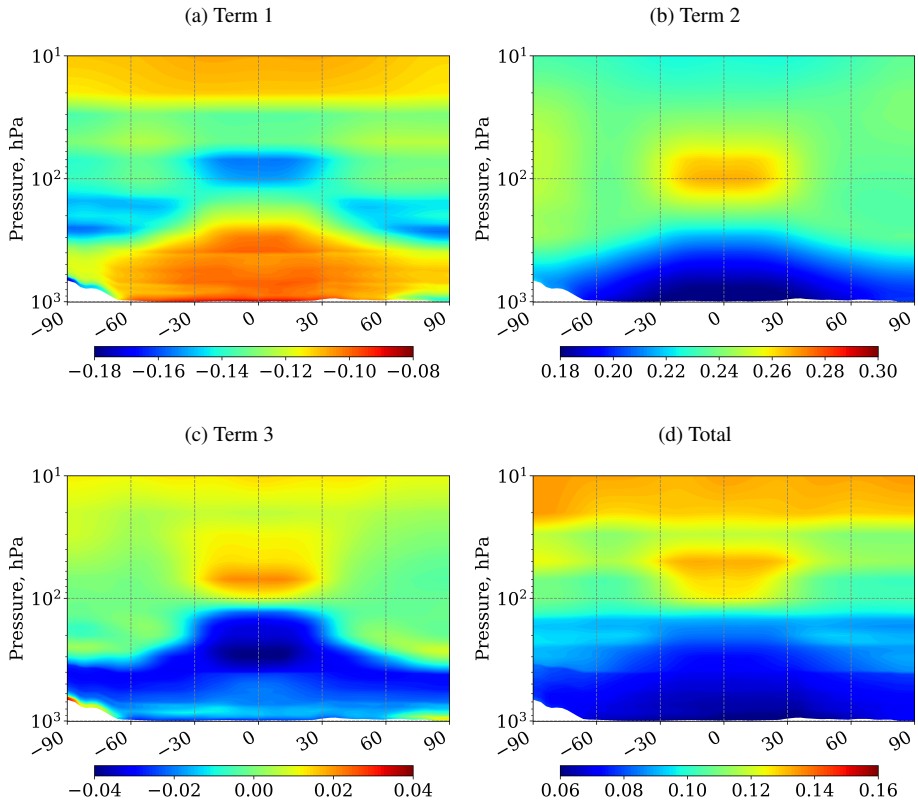

**Figure 2.** Annual averaged (2000–2015) altitude–latitude distributions of diffusion velocity components (a) term 1, (b) term 2, (c) term 3 and (d) total of the three terms [1/km], as described in Eq. 2. These inverse lengths could be converted into equivalent velocities by multiplying by $D_i$ from Eq. 4.

To estimate the contribution of atmospheric conditions to molecular diffusion, we consider the vertical component of the diffusion velocity expressed by the three terms of Eq. 2, following the implementation in SOCRATES. The necessary meteorological fields are taken from the JCDAS reanalysis, and the $^{12}C^{16}O_2$ distribution is calculated by NIES TM.

All three terms undergo a significant change in magnitude and shape near the tropopause (Fig. 2). This emphasizes uncertain-
5  ties in the definition of the tropopause height from the temperature and tracer concentration fields. Around the level of 100 hPa the values are localized in three zones: tropical, northern, and southern. Such a distribution is caused by a strong extra-tropical gradient. The seasonal variation of the sum of the three terms is striking in high-latitude zones with different distributions for the NH summer–fall (JJA–MAM) and winter–spring (DJF–SON) periods due to variation in the BDC (Fig. 3).

Seasonal changes in molecular diffusive flux cause displacement of the position of the maximum and minimum $<\delta>$
10  (Fig. 4). The region of maximum values shifts to the northern boundary of the tropics in NH summer, and to the southern boundary in NH winter. The minimum values are in the Polar Regions. The structure is laminar, with isolines almost along the latitudes. Noticeable distortions are observed over northern Eurasia in the cold season (Fig. 4c). Due to the model grid



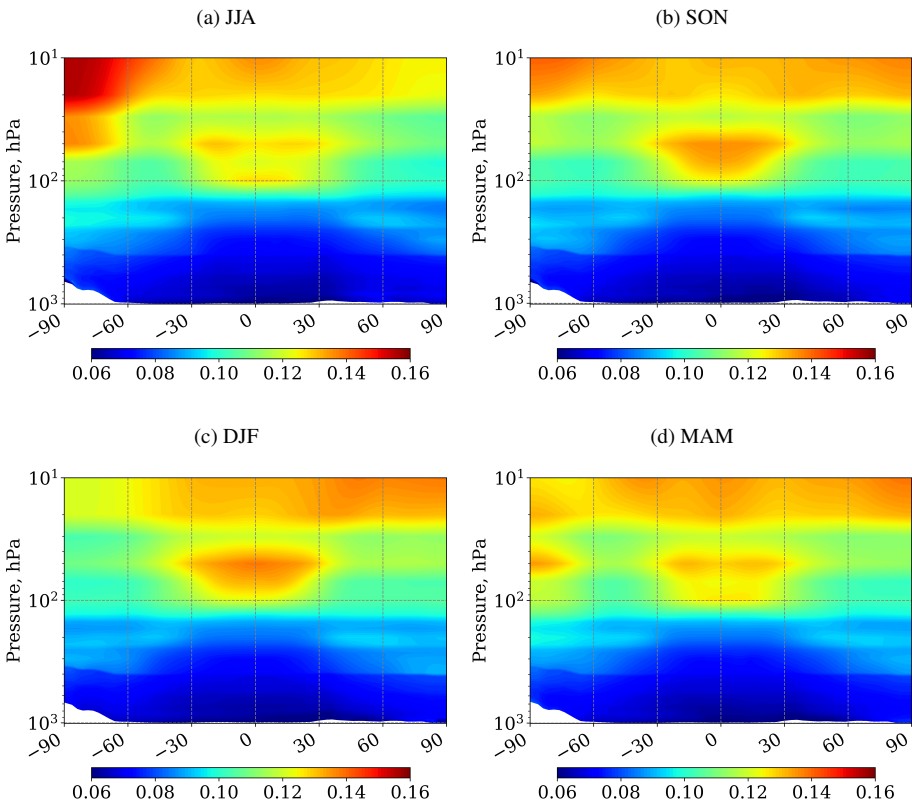

**Figure 3.** Mean altitude–latitude distributions of the sum of the three terms [1/km], as described in Eq. 2 for (a) JJA, (b) SON, (c) DJF, and (d) MAM. The results are averaged for 2000–2015.

**Table 1.** Latitude bands used for averaging $< \delta >$ values.

| Number | Short name | Long name | Latitude interval |
|--------|-----------|-----------|-------------------|
| 1 | SHL | Southern high latitudes | 90°S–60°S |
| 2 | SML | Southern middle latitudes | 60°S–15°S |
| 3 | TPL | Tropical latitudes | 15°S–15°N |
| 4 | NML | Northern middle latitudes | 15°N–60°N |
| 5 | NHL | Northern high latitudes | 60°N–90°N |

construction and the resolution used, the centers in the Polar Regions cannot be resolved well. The shape of isolines with the smallest values is very curved. Thus, the GS process is predominantly two-dimensional with a weak heterogeneity along with longitude.





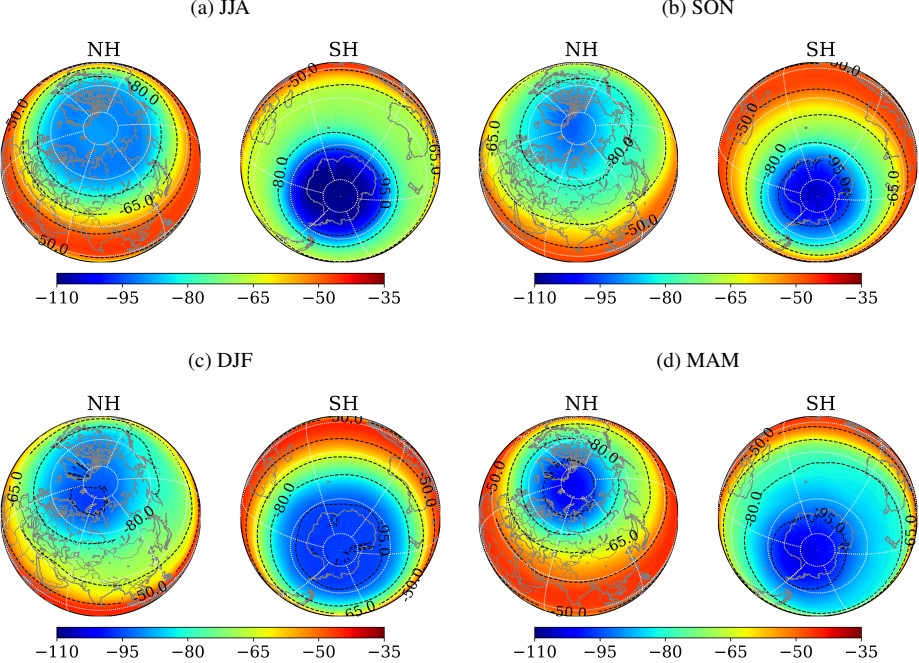

**Figure 4.** Mean latitude–longitude distributions of $< \delta >$ value at the 10 hPa level in South Polar and North Polar projections for (a) JJA, (b) SON, (c) DJF, and (d) MAM. The results are averaged for 2000–2015.

To minimize local temporal and spatial effects, the seasonal variation of vertical profiles was analyzed for five main climate zones: the southern and northern high latitudes, the southern and northern middle latitudes, and the tropics, as shown in Table 1 and averaged over time (2000–2015). It is clear from Fig. 5 that seasonal variation is evident from a level of about 12 km, except in the tropical region, where it starts from 20 km. The amplitude increases with height and reaches a maximum at the top of the model domain. It is quite obvious that seasonal variability is almost imperceptible in the tropics and increases towards the poles. The maximum and minimum values are reached in February and August for the North Pole, and vice versa for the South Pole.

The stronger polar vortex in the SH presumably leads to the enhanced GS (smaller values of $< \delta >$) in JJA in the SH (Fig. 4(a) and Fig. 5(a) light blue line) compared to that in DJF in the NH (Fig. 4(c) and Fig. 5(e)).

Weak sensitivity to seasonal changes of the tracer concentration is a significant advantage of GS over AoA, which is the standard indicator of circulation in the stratosphere. On the other hand, this method requires more accurate sampling tools (i.e., balloon-borne cryogenic samplers) that are more difficult to deploy than other more common methods.





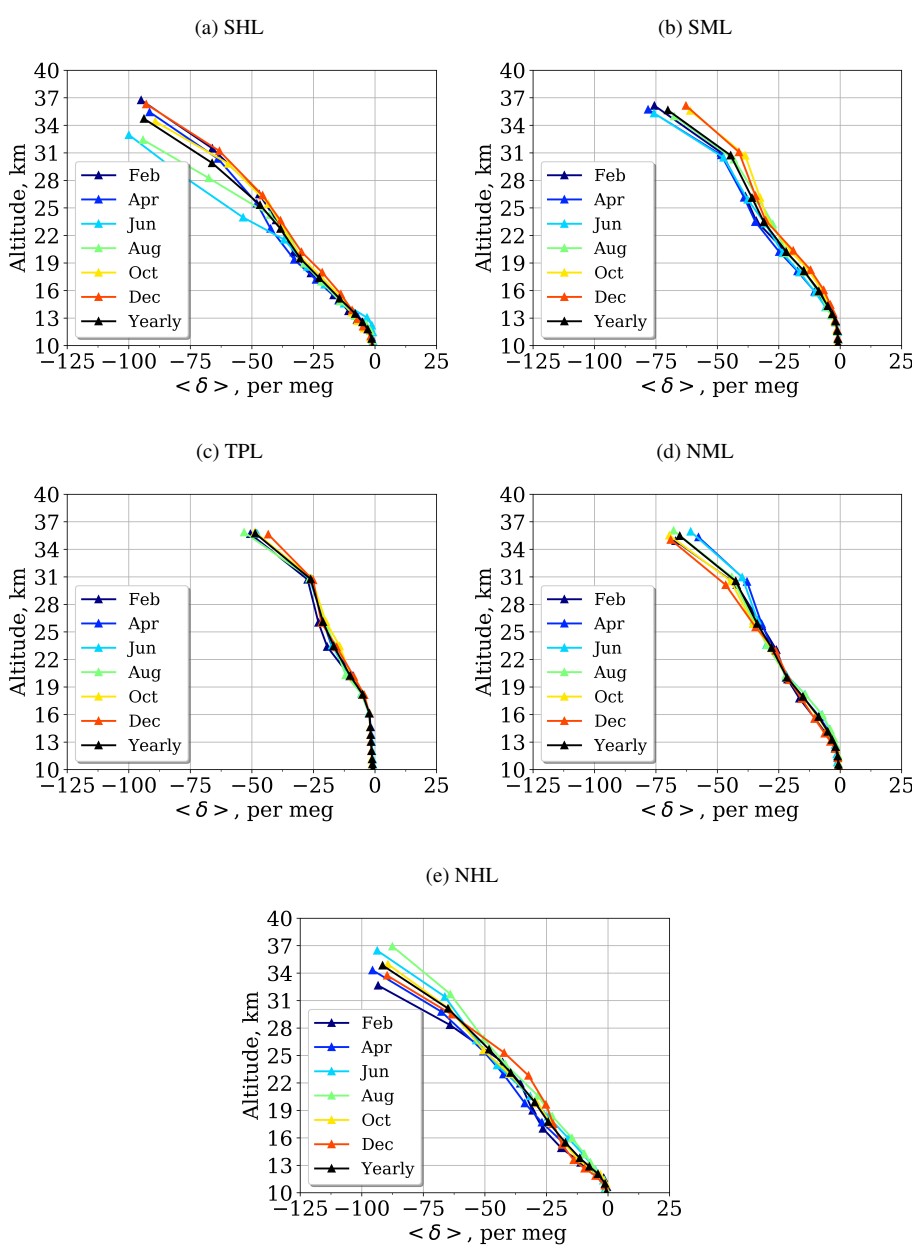

**Figure 5.** Profiles of $<\delta>$ calculated by the model for February, April, June, August, October, December and yearly mean averaged for 2000–2015 over the five latitudinal bands shown in Table 1



**Table 2.** Observation sites.

| Number | Site name | Site coordinates | Observation dates |
|---|---|---|---|
| 1 | Biak, Indonesia | ( 1°S, 136°E) | 22-28 February 2015 |
| 2 | Kiruna, Sweden | (68°N, 21°E) | 18 March 1997 |
| 3 | Sanriku, Japan | (39°N, 142°E) | 8 June 1995, 31 May 1999, 28 August 2000, 30 May 2001, 4 September 2002, 6 September 2004, 3 June 2006, and 4 June 2007 |
| 4 | Syowa, Antarctica | (69°S, 39°E) | 3 January 1998 and 5 January 2004 |
| 5 | Taiki, Japan | (43°N, 143°E) | 22 August 2010 |

## 3.2 Vertical profiles

The modeled results are compared with measurements over the main climatic zones: the circumpolar regions, the temperate latitudes, and the tropical latitudes (Table 2). The collection of stratospheric air samples using a balloon-borne cryogenic sampler was initiated in 1985 at the Sanriku Balloon Center of the Institute of Space and Astronautical Science (Nakazawa

et al., 1995, 2002; Aoki et al., 2003; Goto et al., 2017; Sugawara et al., 2018; Hasebe et al., 2018). The program has continued up to the present. In addition to observations over Japan, stratospheric air samples were also collected over the Scandinavian Peninsula, Antarctica, and Indonesia.

From those air samples, $\delta(^{15}N)$ of $N_2$, $\delta(^{18}O)$ of $O_2$, $\delta(O_2/N_2)$, $\delta(Ar/N_2)$, and $\delta(^{40}Ar)$ were derived to detect GS in the stratosphere (Ishidoya et al., 2006, 2008, 2013). The effect of GS on the isotopic ratio depends on $\Delta m$ rather than on the

atmospheric component, as follows from Eq. 6. Thus $\delta$ values from observations of various tracers and the current simulations can be compared.

Most of the observations were collected in the northern part of Japan over Sanriku (eight profiles) and Taiki (one) in the warm season. Five profiles were observed at the beginning of summer (late May to early June), and four profiles at the end of summer (late August to early September). Typical spring and fall profiles are shown in Fig. 6. For this comparison, the modeled

data are daily output at the nearest grid cell.

In the NH, the tropospheric $CO_2$ is dominated by a strong seasonal cycle due to biospheric activity, which removes $CO_2$ by photosynthesis during the growing phase to reach a minimum in August–September and releases it during boreal autumn and winter with a maximum in April–May. Due to steady growth and seasonal variation, $CO_2$ concentrations in the atmosphere contain both monotonically increasing and periodic signals. Spring profiles are smoother, while in autumn they vary with

height. The AoA shows a strong inversion in the lower part due to seasonal uptake of $CO_2$, as confirmed by CONTRAIL measurements (Machida et al., 2008; Sawa et al., 2008) and the Lagrangian transport model TRACZILLA (Diallo et al., 2017).

For the high latitude sites Kiruna and Syowa (Figs. 7–8) the observed profiles are mainly smooth and have smaller vertical fluctuation, apart from the uppermost level over Syowa station for 5/01/2004.



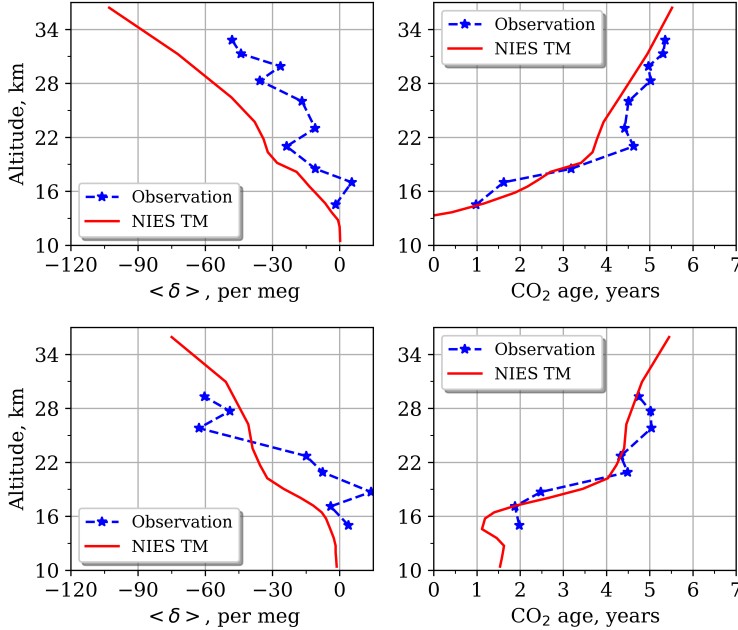

**Figure 6.** Vertical profiles over Sanriku of $< \delta >$ [per meg] (left) and $CO_2$ age [years] (right) calculated with the model (in red) and observed (in blue) for 4 June 2007 (upper panel) and 28 August 2000 (lower panel).

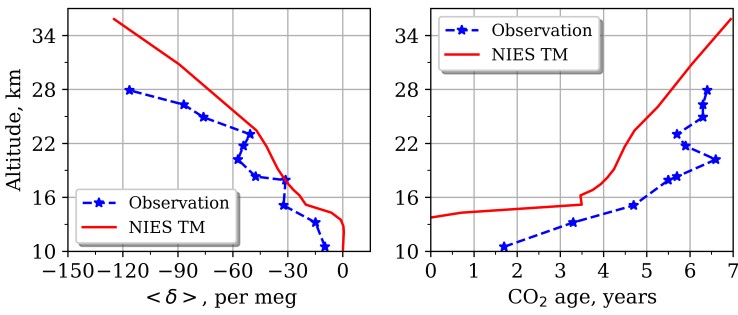

**Figure 7.** Same as Fig. 6, but over the Kiruna site for 18 March 1997.

Due to the limited availability of balloon launching facilities, only one air sampling has been conducted in the equatorial mid-stratosphere over Biak (Hasebe et al., 2018). The observed distribution can be explained by the mixing of large-scale NH and SH background values and long-range transport of $CO_2$ with convective lifting (Inai et al., 2018). For this site, the $< \delta >$ value and the AoA variations with height are very small (Fig. 9), as vertical upwelling pumps young and well-mixed air upward.

Although this is not the model validation paper, it is necessary to evaluate the modeled results by comparison with observations, as the new parameterization for GS simulation was incorporated in NIES TM. A detailed statistical analysis is





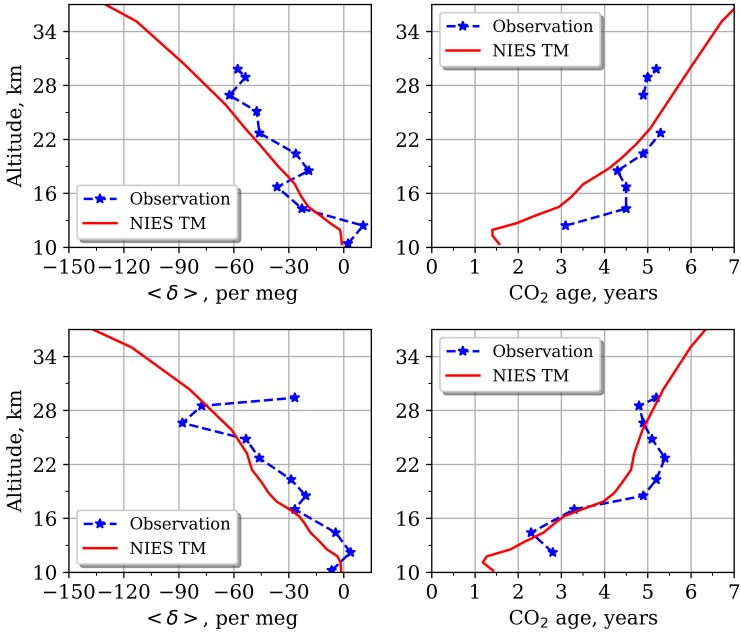

**Figure 8.** Same as Fig. 6, but over Syowa Station for 3 January 1998 (upper panel) and 5 January 2004 (lower panel).

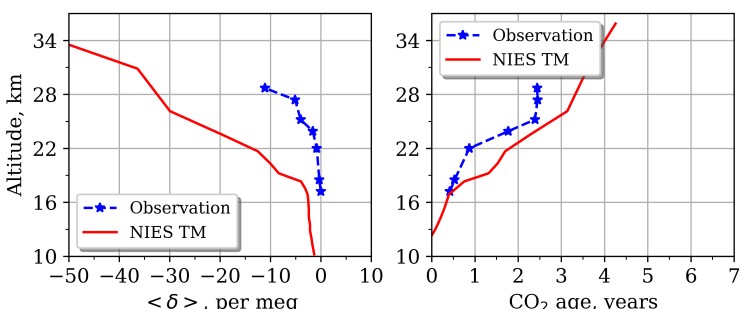

**Figure 9.** Same as Fig. 6, but over the Biak site for 22–28 February 2015.

summarized for the five observational sites in Table 3. This includes standard deviation of the misfit between modeled and observed AoA and $<\delta>$ ($\sigma(\Delta\text{AoA})$ and $\sigma(\Delta<\delta>)$), respectively); the Pearson correlation coefficient between modeled and observed AoA and $<\delta>$ ($r(\text{AoA})$ and $r(<\delta>)$), respectively); as well as the Pearson correlation coefficient between AoA and $<\delta>$ from observation ($r(\text{AoA}, <\delta>)_o$) and from the model ($r(\text{AoA}, <\delta>)_m$). To calculate standard deviation and correlation coefficients only coincident points were selected.

The high values of the correlation coefficients between simulated and observed parameters prove the model efficiency with the implemented parameterization. The lower correlation for GS than for $r(\text{AoA})$ stresses the complexity of the high-precision cryogenic sampling required for GS. For example, most observed profiles show a tendency to have zones of very weak increase





**Table 3.** Standard deviation of misfit between modeled and observed AoA and $< \delta >$ value ($\sigma(\Delta\text{AoA})$ and $\sigma(\Delta < \delta >)$, respectively); the Pearson correlation coefficient between modeled and observed AoA and $< \delta >$ value ($r(\text{AoA})$ and $r(< \delta >)$, respectively); the Pearson correlation coefficient between AoA and $< \delta >$ from observation ($r(\text{AoA}, < \delta >)_o$) and from the model ($r(\text{AoA}, < \delta >)_m$).

| Number | Site name | $\sigma(\Delta\text{AoA})$, yr | $\sigma(\Delta < \delta >)$, per meg | $r(\text{AoA})$ | $r(< \delta >)$ | $r(\text{AoA}, < \delta >)_o$ | $r(\text{AoA}, < \delta >)_m$ |
|---|---|---|---|---|---|---|---|
| 1 | Biak, Indonesia | 0.37 | 9.38 | 0.98 | 0.87 | –0.79 | –0.99 |
| 2 | Kiruna, Sweden | 0.93 | 13.17 | 0.98 | 0.92 | –0.81 | –0.93 |
| 3 | Sanriku, Japan | 0.64 | 15.43 | 0.90 | 0.75 | –0.74 | –0.96 |
| 4 | Syowa, Antarctica | 0.66 | 16.21 | 0.88 | 0.80 | –0.75 | –0.96 |
| 5 | Taiki, Japan | 0.29 | 8.50 | 0.99 | 0.94 | –0.86 | –0.95 |

or even inversion of the parameters starting from a level of 20–25 km. Despite these limitations, the ability to study the physics underlying GS is a fundamental advantage of the 3-D simulation compared with the 2-D simulation as performed by SOCRATES.

The standard deviations $\sigma(\text{AoA})$ and $\sigma(< \delta >)$ quantify model–observation misfits. We stress a tendency of increase towards the high latitudes, although it seems that a larger gap is obtained for Biak. However, if we normalize the standard deviations by the value of the absolute maximum value of the characteristic, the error decreases towards high latitudes.

GS and AoA are useful indicators of atmospheric transport processes. Two other correlations from Table 3 ($r(\text{AoA}, < \delta >)_o$ and $r(\text{AoA}, < \delta >)_m$) quantify their relationship. Since one value increases with height, and the other decreases, the correlation is negative. Modeled results show stronger anti-correlation than that observed, probably due to more straightforward connections in transport simulation; the parametrization used may not take into account additional factors affecting GS. We also do not exclude the influence of observational errors.

### 3.3 Relationship between age of air and $< \delta >$

Further study of the relationship between the AoA and the $< \delta >$ value is useful for understanding the atmospheric processes, as both would be affected to some extent by changes in the stratospheric circulation. For comparison the modeled AoA and $< \delta >$ value were averaged over the same five broad latitude bands as in section 3.1 (see Table 1). Along with the modeled values (solid lines) the observed data are also depicted (symbols) in Fig. 10. Observation sites are quite evenly distributed across the selected latitude bands: Syowa in the southern high latitudes, Biak in the tropics, Sanriku and Taiki in the northern middle latitudes, and Kiruna in the northern high latitudes.

Figure 10 shows a near one-to-one relationship between AoA and GS regardless of latitude and height. The distributions of observations show a similar pattern for the lower part of the stratosphere despite large noise (the positive $< \delta >$ values obtained from observations are not reproducible according to the theory). For layers with an age of 4 and more years, the model–observation discrepancy becomes significant. This indicates that the model tends to underestimate the AoA and overestimate $<$



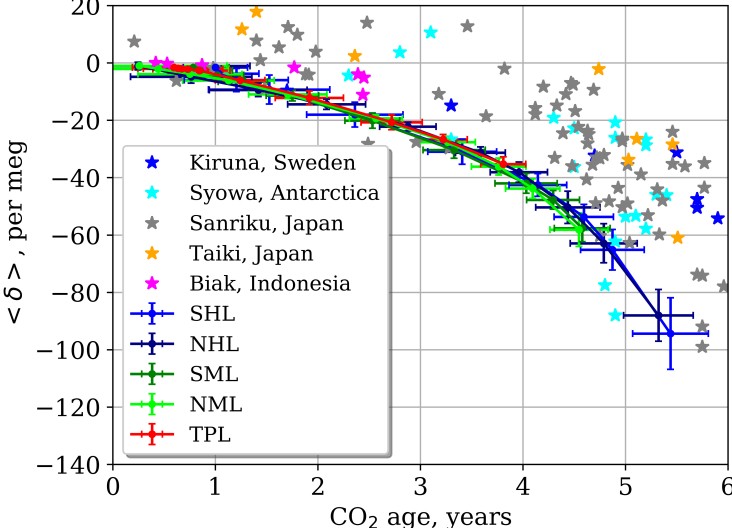

**Figure 10.** Plot of $< \delta >$ value against $CO_2$ age. The modeled results are averaged for the five latitudinal bands shown in Table 1 and over time for 2000–2015. Symbols represent all available observations for Kiruna, Syowa, Sanriku, Taiki, and Biak sites. Error bar values correspond to $1\sigma$.

$\delta >$ in the upper stratosphere. Along with the notable agreement, we must stress some features that emphasize the independence of the driving mechanisms of AoA and GS. As described in section 3.1, the seasonal variation of the tracer concentration has a different effect on AoA and GS. For younger air (0–2 years) the seasonal variability of AoA is maximal and falls with altitude, while the $< \delta >$ variability increases continuously upwards starting from zero.

Sugawara et al. (2018) assumed that vertical upwelling in the tropical tropopause layer (TTL) acts to weaken, and down-welling in high latitudes acts to strengthen the effect on the GS. To minimize the discrepancy between the model-calculated (with SOCRATES) and observed $< \delta >$ values over the equatorial region, a correction of the mean meridional circulation and the horizontal eddy diffusivity was performed. However, that correction cannot fully improve the simulation of the stratospheric circulation and hence the reproducibility of AoA and GS by the SOCRATES model. In this study the tracer simulation is based on reanalysis, which describes the atmospheric circulation in sufficient detail. As part of the global balanced air transport the upwelling and downwelling are limited by various constraints including mass flux conservation. The resulting tracer field distribution simultaneously describes AoA and GS, so both parameters can be used to describe the structure of the global atmospheric circulation. However, the limited set of observations and the limitations of the model do not yet allow us to investigate this mechanism and determine the structure of the AoA–GS relationship in more detail.



## 4 Conclusions

A three-dimensional simulation of GS in the UTLS zone is performed for the first time using NIES TM with a molecular diffusion parameterization. We consider the $<\delta>$ values derived from the distribution of two isotopes $^{12}C^{16}O_2$ and $^{13}C^{16}O_2$. The modeled $<\delta>$ values are compared to observations and the zonal mean distributions from the two-dimensional SOCRATES

model.

In comparison with the SOCRATES simulation, the NIES model has a number of significant advantages: a three-dimensional tracer transport simulation driven by global JCDAS reanalysis and a vertical coordinate with isentropic levels. The model is optimized to run greenhouse gas simulation, as confirmed through various validation and multi-model inter-comparisons. The use of optimized $CO_2$ fluxes provided realistic tracer distribution and seasonality. Along with these strengths, some weaknesses

are also revealed: coarse vertical resolution and the shallow top of the model domain.

The model-to-observation comparison shows that the model with this molecular diffusion parameterization is able to reproduce the mean value and the number of small-scale fluctuations recorded by high-precision cryogenic balloon-borne observations in the lower stratosphere. This reconstruction suggests that the tracer distribution can be explained by the properties of transport, as resolved by meteorological reanalysis and the representation of sub-grid-scale effects as diffusion. Overall, the

implemented molecular diffusion parameterization in NIES TM shows reasonable performance.

We investigated the GS by following the formulation in the SOCRATES model (Ishidoya et al., 2013). Under this framework, the seasonal variations of pressure and temperature gradients are the main contributors that drive seasonal variation in molecular diffusion flux and are thus the cause of variations in $<\delta>$ values.

We found a strong relationship between the modeled GS and AoA, which is the main indicator of circulation in the strato-

sphere. However, in contrast to AoA, the GS has a lower sensitivity to seasonal variability, which is a significant issue in studies of atmospheric circulation. Thus, the modeled GS characteristics provide useful insights and complement AoA information to give a more comprehensive evaluation of structure changes in the UTLS.

*Code and data availability.*   Additional data requests should be addressed to Dmitry Belikov (dmitry.belikov@ees.hokudai.ac.jp).

*Acknowledgements.*   We sincerely thank the balloon engineering group of the Institute of Space and Astronautical Science JAXA for their

cooperation in our stratospheric air sampling. This work is partly supported by the Japan Society for Promotion of Science, Grant-in-Aid for Scientific Research (S) 26220101.





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
