# Peer review of "Three-dimensional simulation of stratospheric gravitational separation using the NIES global atmospheric tracer transport model"

_Atmospheric Chemistry and Physics, 2018_

## Referee Comment (RC1) · Anonymous Referee #1 · 18 Oct 2018

This paper shows results of a CTM with molecular diffusion added to diagnose gravitational separation (GS) in the stratosphere. The different processes that control the diffusive velocity due to GS are discussed and the model distributions are compared to modeled age of air as well as observed profiles of derived GS and mean age. The model theory is clearly described and the comparisons to observations are interesting.

My main concern with the paper is the lack of clear evidence that the GS calculation adds significantly to our understanding of the stratospheric circulation. The conclusion section states that the model GS characteristics provide useful insights into structure changes in the UTLS, particularly over mean age. Mean age from measurements is ill-

defined in the UTLS so it is not used as a measurement-based indicator of circulation changes in that region. There are a number of other trace gases, such as ozone, water vapor and CO, that are commonly used to define the transport and structural changes of the UTLS circulation. Even in the lower stratosphere above 100 hPa, where the seasonal cycle of CO2 can impact mean age estimates, a careful consideration of the boundary conditions can alleviate much of the uncertainty. The simple lag technique used to calculate mean age in this paper is inappropriate for a trace gas with nonlinear growth such as CO2.

The topic is appropriate for ACP and the model simulations of GS are novel. The benefits of these simulations to help interpret stratospheric circulation variability should be more clearly shown and described in my opinion. There are also a number of grammatical errors so I would suggest a more thorough proofreading of the paper is necessary.

Specific comments:

Pg. 2, line 4: add TM abbreviation here since you refer to it on line 7.

Pg. 2, line 23: This is a confusing sentence, the mean meridional circulation is part of the BDC.

Eq. 7: This is an oversimplified way to calculate the mean age from a non-linearly growing trace gas such as CO2. Why not compute the true model mean age using an idealized linearly growing tracer?

Figs 2-4: Very little discussion of these figures in the text. Figure 2 is interesting but the scales are different on each plot and the terms are labeled 1-3 rather than by the physical mechanism responsible for each term so it's difficult to understand what's the important take away. There is only one sentence describing Figure 3, that's not enough. Better to fully discuss the important features of the figure or it should be removed.

Pg. 11, line 20: change "part" to "altitudes"

Pg. 12, line 1: change "sampling" to "sample"

Pg. 12, line 6: change "the model" to "a model"

Pg. 16, lines 20-22: As mentioned above, these statements aren't necessarily supported in the paper.

---

## Referee Comment (RC2) · Anonymous Referee #2 · 23 Oct 2018

The authors conducted numerical simulation of carbon dioxide isotopes, 12C16O2 and 13C16O2 by using the NIES global tracer transport model (CTM), and clarified its gravitational separation (GS) due to the difference in the molecular diffusion. The CTM, which is an offline 3-D passive tracer transport model based on isentropic vertical coordinates, was driven by the reanalysis dataset, JMA Climate Data Assimilation System (JCDAS). Results of the 3-D CTM were found to be much more realistic than the results of a 2-D transport model. The CTM showed the GS apparently increasing with increasing altitude and latitude in the stratosphere, and also suggested a unique relationship of GS with the age of air (AoA). This work made an important progress in understanding of the 3-D distribution of GS in the stratosphere. However, there are many incomplete

discussions, particularly of physical interpretations of simulated results. I recommend to publish the manuscript in ACP after addressing the following comments.

1. Generally speaking, the turbulent eddy diffusion is much greater than the molecular diffusion in the troposphere. The eddy vertical diffusions may be enhanced by the large wave activity in the extratropical stratosphere. The authors should describe how the CTM treats the eddy vertical diffusion in the troposphere and stratosphere, and discuss how they affect the GS distributions in the stratosphere.

2. The CTM adopts isentropic vertical coordinates, where the diabatic heating assessed in the reanalysis is used to estimate the vertical velocity. I think the choice of isentropic coordinates is adequate, because its vertical motions are free from the gravity wave noises. Note that conventional pressure coordinates tend to overly express the vertical mixing due to gravity wave noises. A problem in this work is that the vertical velocity was assessed from the seasonal mean diabatic heating. It neglected the short-term temporal variation of actual instantaneous diabatic heating, and underestimated their contributions to the vertical diabatic mixing.

3. The characteristics of Brewer-Dobson circulation in a reanalysis have been recognized to vary significantly depending on the reanalysis, and to be subject to systematic errors of the reanalysis. In this experiment, the CTM was driven by using a reanalysis. The systematic errors of reanalysis may degrade the simulated GS. How do the authors think about this problem?

4. Figure 4 showed that the geographical distribution of $<\delta>$ value is significantly different between the northern and southern hemispheres. According to the authors, the stronger polar vortex enhances the GS in JJA-SH compared with DJF-NH. Furthermore, the GS differences may be caused by the Brewer-Dobson circulation and horizontal diffusions on isentropic surfaces. The authors should clarify major mechanisms causing the actual differences in GS distributions.

5. Figure 10 showed that the $<\delta>$ value decreases very rapidly after the age exceeding

4 years. It means that the GS is highly nonlinear to the residence time of "Age" in the stratosphere. We would like to know the mechanisms for the GS acceleration in layers with an age of 4 or more years in the model.

---

## Author Comment (AC1) · 12 Feb 2019

The comment was uploaded in the form of a supplement:
https://www.atmos-chem-phys-discuss.net/acp-2018-835/acp-2018-835-AC1-supplement.zip

---

## Author Comment (AC2) · 12 Feb 2019

The comment was uploaded in the form of a supplement:
https://www.atmos-chem-phys-discuss.net/acp-2018-835/acp-2018-835-AC2-supplement.zip

---

## Author Response (AR1)

We thank the reviewers for their constructive and helpful suggestions. We have provided our responses to the reviewers' comments and believe that our manuscript is much improved as a result.

The main paper improvements are:

1. The AoA analysis was revised. We used the idealized linearly growing "surface" tracer proposed in the Age of air intercomparison project (Krol et al., 2018). Therefore, all figures that depict AoA were updated: the zonal mean plot and the comparing with observations (Figures 1b, 5-8).
2. Figure 2 was removed.
3. The script for calculating averages for 3D distributions was revised, therefore Fig. 1a and Fig. 3 were updated.
4. The script for calculating seasonal averages was revised, therefore panels at Fig. 2 (the diffusion velocity of SON and MAM are exchanged) and Fig. 4 (less spread, large values around the level of 35km) were updated.
5. Due to scripts revision, inaccuracies in figures were eliminated. Thus, consistency between vertical profiles and 2D distributions has improved. All figures show $<\delta>$ values around -100 per meg in the high-latitudes, -70 per meg in the middle-latitudes and -50 per meg in tropics.

The reviewers' specific comments (shown in blue) are addressed below.

**Anonymous Referee #1**

This paper shows results of a CTM with molecular diffusion added to diagnose gravitational separation (GS) in the stratosphere. The different processes that control the diffusive velocity due to GS are discussed and the model distributions are compared to modeled age of air as well as observed profiles of derived GS and mean age. The model theory is clearly described and the comparisons to observations are interesting.

C1:

My main concern with the paper is the lack of clear evidence that the GS calculation adds significantly to our understanding of the stratospheric circulation. The conclusion section states that the model GS characteristics provide useful insights into structure changes in the UTLS, particularly over mean age.

Please see C3.

C2:

Mean age from measurements is ill-defined in the UTLS so it is not used as a measurement-based indicator of circulation changes in that region. There are a number of other trace gases, such as ozone, water vapor and CO, that are commonly used to define the transport and structural changes of the UTLS circulation. Even in the lower stratosphere above 100 hPa, where the seasonal cycle of $CO_2$ can impact mean age estimates, a careful consideration of the boundary conditions can alleviate much of the uncertainty. The simple lag technique used to calculate mean age in this paper is inappropriate for a trace gas with nonlinear growth such as $CO_2$.

The mean age calculation method was updated. For details please see C7.

C3:

The topic is appropriate for ACP and the model simulations of GS are novel. The benefits of these simulations to help interpret stratospheric circulation variability should be more clearly shown and described in my opinion.

As mentioned above, this work is a continuation and expansion of earlier research. So we summarized the limitations that need to be overcome and those properties that should be studied in future works **(p.16, l.19–23)**: "However, due to the simplified approach and parameterizations, the presented simulation of the GS using the NIES model could not fully achieve the potential of 3D modeling. Modern reanalysis dataset and recently developed transport models effectively simulated the upper atmosphere can be employed to address

these issues. Since this work is the first in 3D modeling of GS, we believe this insight is useful for the scientific community working in the field of the UTLS studies."

There are also a number of grammatical errors so I would suggest a more thorough proofreading of the paper is necessary.

English editing company managed by native speakers rechecked English grammar in the original manuscript and didn't find any other errors except shown below. Therefore, if errors are indicated, we will be grateful.

Specific comments:

C5:

Pg. 2, line 4: add TM abbreviation here since you refer to it on line 7.

Done.

C6:

Pg. 2, line 23: This is a confusing sentence, the mean meridional circulation is part of the BDC.

Revised as: "is perhaps best known for being a proxy for the rate of the stratospheric mean meridional circulation and the whole BDC."

C7:

Eq. 7: This is an oversimplified way to calculate the mean age from a non-linearly growing trace gas such as $CO_2$. Why not compute the true model mean age using an idealized linearly growing tracer?

The age of air calculation method was updated. The text was revised **(p.6, l.10–13)**: "Along with the $<\delta>$ value we analyze the AoA (Fig. 1). For this, we used the idealized linearly growing "surface" tracer proposed in the Age of air intercomparison project (Krol et al., 2018). To fit with our analysis period we extended the original simulation period (1988-2014) to 29 years (1988–2016) with a shorter (10 years) spin-up, as less time required to reach equilibrium for the AoA analysis."

C8:

Figs 2-4: Very little discussion of these figures in the text. Figure 2 is interesting but the scales are different on each plot and the terms are labeled 1-3 rather than by the physical

mechanism responsible for each term so it's difficult to understand what's the important take away. There is only one sentence describing Figure 3, that's not enough. Better to fully discuss the important features of the figure or it should be removed.

The scale of the figures is quite diverse, so a selection of common color bar is complicated.

Figure 2 was removed.

Discussion of the Figure 3 (now Figure 2) was revised **(p.7, l.1–10)**: "To estimate the contribution of atmospheric conditions to molecular diffusion, we consider the sum of the three terms in the square bracket of Eq. 5. Because the contribution of the first term (concentration gradient) is relatively small, the second term (originated from pressure gradient in Eq. 1) is the major contributor among the three terms. Therefore, sum of three terms can be approximated by the difference between the reciprocals of two scale heights (hereafter referred to as $L_i^{-1}$). It has a dimension reciprocal to the length, and is interpreted as a measure of the efficiency of vertical molecular diffusion under gravity. In view of the essentially one-dimensional nature of GS, it is interesting to consider how $L_i^{-1}$ distributes in the troposphere and stratosphere. Figure 2 shows the latitude-height distribution of $L_i^{-1}$ averaged in each season for the case of $^{12}C^{16}O_2$. Here the positive values indicate that $^{12}C^{16}O_2$ molecules descend relative to major constituents. The temperature fields necessary for the calculation are taken from the JCDAS reanalysis. Since $L_i^{-1}$ is in inverse proportion to temperature, it is generally small in the troposphere and takes maxima in the cold region such as the tropical tropopause region and the winter time stratosphere."

C9:

Pg. 11, line 20: change "part" to "altitudes

Done

C10:

Pg. 12, line 1: change "sampling" to "sample"

Done

C11:

Pg. 12, line 6: change "the model" to "a model"

Done

C12:

Pg. 16, lines 20-22: As mentioned above, these statements aren't necessarily supported in the paper.

See C3

**Anonymous Referee #2**

The authors conducted numerical simulation of carbon dioxide isotopes, $^{12}C_{16}O_2$ and $^{13}C_{16}O_2$ by using the NIES global tracer transport model (CTM), and clarified its gravitational separation (GS) due to the difference in the molecular diffusion. The CTM, which is an offline 3-D passive tracer transport model based on isentropic vertical coordinates, was driven by the reanalysis dataset, JMA Climate Data Assimilation System (JCDAS). Results of the 3-D CTM were found to be much more realistic than the results of a 2-D transport model. The CTM showed the GS apparently increasing with increasing altitude and latitude in the stratosphere, and also suggested a unique relationship of GS with the age of air (AoA). This work made an important progress in understanding of the 3-D distribution of GS in the stratosphere. However, there are many incomplete discussions, particularly of physical interpretations of simulated results. I recommend to publish the manuscript in ACP after addressing the following comments.

C1:

Generally speaking, the turbulent eddy diffusion is much greater than the molecular diffusion in the troposphere. The eddy vertical diffusions may be enhanced by the large wave activity in the extratropical stratosphere. The authors should describe how the CTM treats the eddy vertical diffusion in the troposphere and stratosphere, and discuss how they affect the GS distributions in the stratosphere.

The eddy vertical diffusion in the troposphere is described by Belikov et al. (2013a): The parametrization of turbulent diffusivity follows the approach used by Hack et al. (1993), with transport processes in the planetary boundary layer (PBL) and free troposphere evaluated separately. Turbulent diffusivity above the top of the PBL is calculated from local stability as a function of the Richardson number and is set to a constant value of 40 m2 s−1 under an assumption of well-mixed air below the PBL top. Three-hourly PBL heights are taken from the European Centre for Medium-Range Weather Forecasts (ECMWF) ERA-Interim Reanalysis.

*Hack, J. J., Boville, B. A., Briegleb, B. P., Kiehl, J. T., Rasch, P. J., and Williamson, D. L.: Description of the NCAR community climate model (CCM2), NCAR/TN-382, 108, 1993.*

Added **(p.5, l.7−11)**: "The eddy vertical diffusion in the stratosphere is often neglected in CTMs. However, it should be considered along with molecular diffusion here. The turbulent diffusion coefficient is estimated from parameterizations of gravity wave dissipation (Lindzen et al., 1981) similar to the SOCRATES model. In general, the eddy diffusion mixes concentrations in the volume, reduces vertical stratification and thereby weakens the molecular diffusion effect, as discussed by Kockarts et al. (2002)."

*Lindzen, R. S.: Turbulence and stress owing to gravity wave and tidal breakdown, J. Geophys. Res., 86, 9707–9714, 1981.*

C2:

The CTM adopts isentropic vertical coordinates, where the diabatic heating assessed in the reanalysis is used to estimate the vertical velocity. I think the choice of isentropic coordinates is adequate, because its vertical motions are free from the gravity wave noises. Note that conventional pressure coordinates tend to overly express the vertical mixing due to gravity wave noises. A problem in this work is that the vertical velocity was assessed from the seasonal mean diabatic heating. It neglected the short-term temporal variation of actual instantaneous diabatic heating, and underestimated their contributions to the vertical diabatic mixing.

The NIES model was developed to simulate greenhouse gases in near-surface layers, so the priority was to reproduce processes in the troposphere (seasonal cycle, inter-hemispheric gradient, moisture convection, etc.). Modeling of the stratosphere was tuned on the basis of the global balance of the tracer and the reproduction of AoA (Belikov et al., 2011, 2013a,b). Climatological heating rate meets these requirements and reduced nosy perturbations in the stratosphere. Although a certain proportion of short-scale changes may be lost in this case, however, the vertical profiles of the parameters studied are reproduced quite confidently as shown in Table 3.

C3:

The characteristics of Brewer-Dobson circulation in a reanalysis have been recognized to vary significantly depending on the reanalysis, and to be subject to systematic errors of the reanalysis. In this experiment, the CTM was driven by using a reanalysis. The systematic errors of reanalysis may degrade the simulated GS. How do the authors think about this problem?

Indeed, due to the features (grid type, horizontal and vertical resolutions, dynamical core, advection algorithm and etc.) modern reanalysis show very different performance in reproduction of the BDC characteristics.

The following sentence added **(p.14, l.32 – p.15, l.2)**: "Chabrillat et al. (2018) presented a consistent intercomparison of AoA according to five modern reanalyses (ERA-Interim, JRA-55, MERRA, MERRA-2 and CFSR) and found significant diversity in the distributions which were obtained with BASCOE transport model, depending on the input reanalysis. They have also found large disagreement between the five reanalyses with respect to the long-term trends of AoA. Thus, an ambitious multi-reanalyses approach is needed to distinguish what is robust in the current GS results from what is not."

C4:

Figure 4 showed that the geographical distribution of <δ> value is significantly different between the northern and southern hemispheres. According to the authors, the stronger polar vortex enhances the GS in JJA-SH compared with DJF-NH. Furthermore, the GS differences may be caused by the Brewer-Dobson circulation and horizontal diffusions on isentropic surfaces. The authors should clarify major mechanisms causing the actual differences in GS distributions.

Discussion of the Figure 4 (now Figure 3) was revised **(p.7, l.11 – p.8, l.5)**: "The enhancement of $Li^{-1}$ does not readily result in a remarkable GS, because it is the difference of $Li^{-1}$ between $13C16O_2$ and $12C16O_2$ that creates GS in our case. For all that, we could expect that the enhancement of $Li^{-1}$ combined with the long stratospheric transit time in the polar stratosphere will be favorable for GS. Figure 3 compares the horizontal distributions of the seasonal mean $< \delta >$ on 10 hPa pressure surface in polar projections. We can see remarkable GS (small values of $< \delta >$) in both Polar Regions exhibiting surprisingly clear axial symmetry. In the present analysis, the physical processes that drive GS (Eq. 1) have been rearranged in the form of Eq. 5 to separate the contribution to GS in two factors, one the concentration gradient (the first term) and the other the temperature structure. A stronger seasonal variability of GS in the southern hemisphere is caused by changes in vertical pressure gradient (Eq. 1) reflected to those in scale height difference between species."

C5:

Figure 10 showed that the <δ> value decreases very rapidly after the age exceeding 4 years. It means that the GS is highly nonlinear to the residence time of "Age" in the stratosphere. We would like to know the mechanisms for the GS acceleration in layers with an age of 4 or more years in the model.

The following section is added **(p.14, l.13–15)**: "The $< \delta >$ value decreases very rapidly after the age exceeding 4 years, as the molecular diffusion coefficient increases with increasing height due to its pressure dependence (Eq. 4), which causes the enhancement of gravitational separation with increasing height. The mechanism does not affect AoA significantly, in the stratosphere (Ishidoya et al., 2008, 2013; Sugawara et al., 2018). This emphasizes a nonlinearity in the GS-AoA relationship in the stratosphere."